# Learning Parsimonious Deep Feed-forward Networks

## Abstract

Convolutional neural networks and recurrent neural networks are designed with network structures well suited to the nature of spacial and sequential data respectively. However, the structure of standard feed-forward neural networks (FNNs) is simply a stack of fully connected layers, regardless of the feature correlations in data. In addition, the number of layers and the number of neurons are manually tuned on validation data, which is time-consuming and may lead to suboptimal networks. In this paper, we propose an unsupervised structure learning method for learning *parsimonious* deep FNNs. Our method determines the number of layers, the number of neurons at each layer, and the sparse connectivity between adjacent layers automatically from data. The resulting models are called *Backbone-Skippath Neural Networks* (BSNNs). Experiments on 17 tasks show that, in comparison with FNNs, BSNNs can achieve better or comparable classification performance with much fewer parameters. The interpretability of BSNNs is also shown to be better than that of FNNs.

## 1 Introduction

Deep neural networks have made breakthroughs in all kinds of machine learning tasks (LeCun et al., 2015; Hinton et al., 2012a; Mikolov et al., 2011), specifically with convolutional neural networks (CNNs) for tasks with spacial data (Krizhevsky et al., 2012) and recurrent neural networks (RNNs) for tasks with sequential data (Sutskever et al., 2014). One of the key reasons for the effectiveness of CNNs and RNNs is the well-designed network structures together with the parameter sharing schemes. For example, in the convolution layers of CNNs, each neuron is connected to a local region in the input volume instead of all the input neurons. Besides, the neurons in the same channel share the same set of weights. This design utilizes the local and "stationary" properties of spacial data and consequently forms effective feature extractors. In addition, it also prevents CNNs from having an exploding number of parameters when the networks become deeper and deeper.

However, in practice, there are also many data which are neither spacial nor sequential, and hence the only applicable neural networks are the standard feed-forward neural networks (FNNs). In contrast to CNN and RNN, FNN's network structure is simple. It consists of multiple layers of neurons and each layer is fully connected to the next layer up, without considering any correlations in data or among neurons. The network structure has two main shortcomings. The first is that, there can be high connection redundancies. As the number of layers and the number of neuron at each layer increase, the number of parameters increases quickly, which can cause severe overfitting. The other shortcoming is that, ignoring all the correlations existing in data weakens the model's strength (as a feature extractor) and hurts the model's interpretability.

We are interested in learning *parsimonious* deep feed-forward neural networks. The goal is to learn FNNs which contain as few parameters as possible. Parsimonious FNNs are desirable for several reasons. Firstly, fewer parameters can ease overfitting. Secondly, parsimonious FNNs require less storage and computation than FNNs, which makes it possible to be run on devices like mobile phones. Lastly, parsimonious FNNs can have very flexible and different structures from each other depending on the specific tasks and data. This would help the models fit the data well and also have good interpretability. In general, it is desirable to solve a problem using the simplest model possible because it implies a good understanding of the problem.

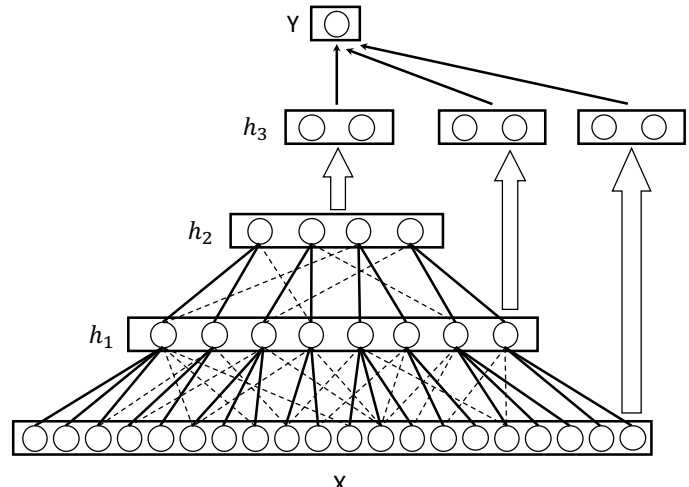

Figure 1: Model structure of Backbone-Skippath Neural Network. The wide layers with sparse connections ($x - h_1$, $h_1 - h_2$) form the Backbone path. The narrow fully-connected layers ($x - h_3$, $h_1 - h_3$, $h_2 - h_3$) are the Skip-paths. The number of units at $h_3$ is relatively smaller than that at $x$, $h_1$ and $h_2$.

Learning parsimonious FNNs is challenging mainly because we need to determine the sparse connectivity between layers. Network pruning is a potential way to achieve this. However, it requires to start from a network which is much larger than necessary for the task at hand. This can cause a lot of computations wasted on those useless connections. In addition, network pruning is not able to learn the number of units and number of layers.

In this paper, we assume that data are generated by a sparse probabilistic model with multiple layers of latent variables, and view the feed-forward network to be built as a way to approximate the relationships between the observed variables and the top-level latent variables in the probabilistic model. The level 1 latent variables induce correlations among the observed variables. Therefore, it is possible to determine them by analysing how the observed variables are correlated. Similarly, by analysing how the level 1 latent variables are correlated, we can determine the level 2 latent variables, and so on. We empirically show that our method can significantly reduce the number of parameters in FNNs, and the resulting model still achieves better or comparable results than FNNs in 17 classification tasks.

## 2 RELATED WORKS

**Network Structure Learning**  One early attempt to learn network structure for FNNs is the approach based on constructive algorithms (Ash, 1989; Bello, 1992; Kwok & Yeung, 1997). These algorithms start from a small network and gradually add new neurons to the network until some stopping criterion are met (e.g. no more performance gain is observed). They require manually-designed strategies to decide how to connect new neurons to the existing network. Besides, each time when new neurons are introduced, the network needs to be retrained completely or partially. Lately, Adams et al. (2010) proposes to learn the structure of deep belief networks by using cascading Indian buffet process, which is very time-consuming. In Chen et al. (2017b), the authors propose a structure learning method, based on hierarchical latent tree analysis (Liu et al., 2014; Chen et al., 2016; 2017a), for RBM-like models. The method automatically determines the number of hidden units and the sparse connections between layers. However, it is not tested on deep models and in supervised learning tasks. Recently, reinforcement learning (Baker et al., 2017; Zoph & Le, 2017) and genetic algorithms (Real et al., 2017; Xie & Yuille, 2017) are also applied to learning complex structures for CNNs. Generally, these methods require tens of thousands of full training runs before giving a feasible network structure, which is prohibitive for many applications.

**Network Pruning**   In contrast to constructive algorithms, network pruning starts from a large network and prune connections or neurons to achieve structure learning. Optimal Brain Damage (Cun et al., 1990) and Optimal Brain Surgeon (Hassibi et al., 1993) prune connections based on the Hessian matrix of the loss function. Recently, Han et al. (2015) proposes to conduct pruning by iteratively pruning connections with absolute weight value smaller than a threshold and retraining the network. One drawback of the method is that the retraining process is time-consuming. Guo et al. (2016) proposes Dynamic Network Surgery which conducts parameter learning and connection pruning simultaneously and avoids the retraining process. Moreover, it also allows mistakenly pruned connections to be rebuilt in subsequent training. Similar to connection pruning, neurons pruning methods are proposed and tested in Srinivas & Babu (2015); Li et al. (2017). The main drawback of all these pruning methods is that, they require to start from a network which is larger than necessary for the task at hand. This causes some wasted computations on the useless connections or neurons. In addition, the number of layers is still set manually instead of learned from data.

## 3   METHODS

In this section, we present a method for learning parsimonious deep FNNs. The method is called *Parsimonious Structure Analysis (PSA)*. PSA learns a model which contains two parts as shown in Figure 1. The first is the main part of the model, called the *Backbone*. It is a wide, deep but sparse feed-forward path in the network. The second part is the *Skip-paths*. It consists of multiple narrow paths, each of which is a fully-connected layer. We call the resulting model *Backbone-Skippath Neural Network (BSNN)*. We will introduce how PSA learns the Backbone and the Skip-paths in Section 3.1 and Section 3.2 respectively.

### 3.1   LEARNING THE BACKBONE

Structure learning for neural networks is challenging since generally the features in data do not always have apparent relationships as the units in convolutional networks. In a convolutional layer, units in a feature map are only connected to a group of units strongly correlated in the spacial dimension at the layer below. This significantly reduces the number of parameters in CNNs and is essential if we want to learn a very sparse structure. The same intuition can be applied to general data other than images in feed-forward neural networks. A hidden unit, detecting one particular feature such as co-occurrence pattern, should only be connected to a group of units that are strongly correlated in the layer below. However, unlike CNNs where the spatial correlation is apparent, the correlations of units in feed-forward neural networks are not easy to discover. In PSA, we propose to apply *Hierarchical Latent Tree Analysis (HLTA)* (Liu et al., 2014; Chen et al., 2016; 2017a) to identify the co-occurrence patterns among units and construct hidden units to explain the co-occurrence patterns.

#### 3.1.1   LEARNING A TWO-LAYER STRUCTURE

PSA treats the input features as a set of isolated random variables as in Figure 3(a). Although no apparent spacial or sequential relationships exist among the variables, PSA seeks to discover the correlations among the variables and groups the highly correlated ones together. It starts from finding two most correlated variables to form one group and keeps expanding the group if necessary. Let $S$ denotes the set of observed variables which haven't been included into any variable groups.

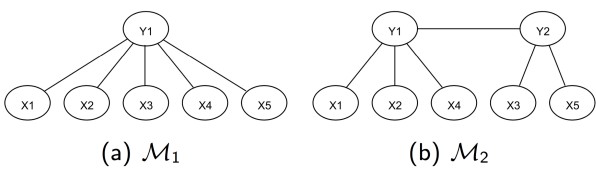

(a) $\mathcal{M}_1$          (b) $\mathcal{M}_2$

Figure 2: Example: (a) The best model with one latent variable for five observed variables. (b) The best model with two latent variables for five observed variables.

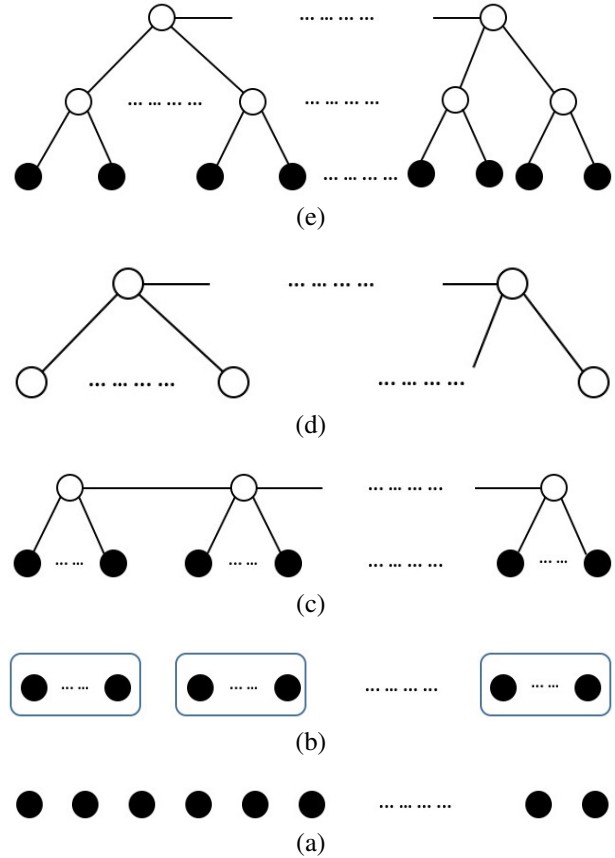

Figure 3: The structure learning steps of PSA. Black circles represent observed variables while white circles represent latent variables. (a) A set of observed variables. (b) Partitions the observed variables into groups. (c) Introduces a latent variable for each group and link the latent variables up as a Chow-Liu tree. (d) Converts the latent variables at layer 1 into observed variables and repeat the previous process on them. (e) Stacks the layer 2 latent variables on the top previous model.

PSA firstly computes the mutual information between each pair of observed variables. Then it picks the pair in $S$ with the highest mutual information and uses them as the seeds of a new variable group $G$. New variables from $S$ are then added to $G$ one by one in descending order of their mutual information with variables already in $G$. Each time when a new variable is added into $G$, PSA builds two models ($\mathcal{M}_1$ and $\mathcal{M}_2$) with $G$ as the observed variables. The two models are the best models with one single latent variable and two latent variables respectively, as shown in Figure 2. PSA computes the BIC scores of the two models and tests whether the following condition is met:

$$BIC(\mathcal{M}_2|D) - BIC(\mathcal{M}_1|D) \leq \delta,$$

where $D$ is the dataset and $\delta$ is a threshold which is usually set at 3 (Chen et al., 2017a). When the condition is met, the two latent variable model $\mathcal{M}_2$ is not significantly better than the one latent variable model $\mathcal{M}_1$. Correlations among variables in G are still well modeled using a single latent variable. Then PSA keeps on adding new variables to $G$. If the test fails, PSA takes the subtree in $\mathcal{M}_2$ which doesn't contain the newly added variable and identifies the observed variables in it as a finalized variable group. The group is then removed from $S$. And the above process is repeated on $S$ until all the variables in $S$ are partitioned into disjoint groups. An efficient algorithm progressive EM (Chen et al., 2016) is used to estimate the parameters in $\mathcal{M}_1$ and $\mathcal{M}_2$.

As shown in Figure 2(b), after the above process, all the observed variables are partitioned into disjoint groups such that the variables in each group are strongly correlated and their correlations can be explained using a single latent variables. Then PSA introduces a latent variable for each group and computes the mutual information among the latent variables. After that, it links up the

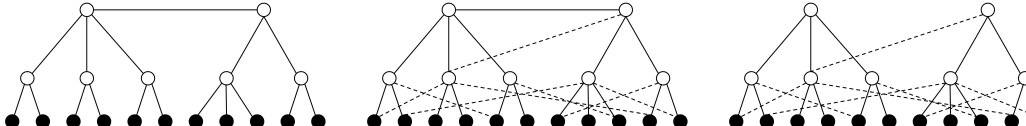

Figure 4: Expanding the tree structure for the Backbone path: A three-layer structure is first learned (left). New connections are added to all the layers according to empirical conditional mutual information (middle). The connections between variables at the top layer are removed and the structure is finalized (right).

latent variables to form a Chow-Liu tree (Chow & Liu, 1968). The result is a latent tree model (Pearl, 1988; Zhang, 2004), as shown in Figure 2(c). Parameter estimation for the model is done using the EM algorithm. Since the model is tree-structured, EM is efficient in this process.

### 3.1.2 LEARNING A DEEP STRUCTURE

While the above procedure gives us a one-layer network, we seek to build deep model to capture the long-range correlations among variables. We perform the construction of deep structure in a layer-wise manner. Using the obtained one-layer model, PSA converts the latent variables into observed ones through data completion. With this, another layer of latent variables can be learned in the same manner as the first layer by grouping the first-layer latent variables and linking up the groups, as in Figure 2(d). Then the two models can be stacked up to form a three-layer network, with the latent variables in the higher layer capturing longer-range correlations of the observed variables. This procedure can be recursively conducted to build deep hierarchy until the number of variables at the top layer falls below a threshold $K$. And it results in a hierarchical latent tree model (Liu et al., 2014; Chen et al., 2016; 2017a).

### 3.1.3 EXPANDING TREE STRUCTURE

While the above deep structure captures the most important correlations among the observed variables, the tree structure might cause underfitting for discovering non-trivial correlations. Thus we introduce additional links to model the salient interactions that are not captured by the tree model. For each latent variable $V_l$ at level $l$, PSA considers adding connections to link it to more nodes at level $l-1$. To do so, PSA considers how closely $V_l$ is related to each node $V_{l-1}$ at level $l-1$ given the parent variable $Z$ of $V_{l-1}$. The strength of correlation is measured using the conditional mutual information:

$$I(V_l, V_{l-1}|Z).$$

The top $N$ nodes with the highest $I(V_l, V_{l-1}|Z)$ are then connected to $V_l$. After expanding the connections for all the layers, PSA removes the links among the variables at the top layer and uses the resulting structure for the Backbone. The process of expanding tree structure is illustrated in Figure 4.

### 3.2 SKIP-PATHS

Although the Backbone path is deep and wide, its sparsity can easily lead to model which cannot capture global features. For example, suppose there is an essential feature which is correlated to all the input features. When the Backbone path is very sparse, even after multiple layers of projections, it is still unlikely that there will be a feature in the model which is projected from all the input features.

To tackle the above problem, we introduce Skip-paths to our BSNN. Figure 1 shows the whole model structure of BSNN. The path from $x$ to $h_2$ illustrates the Backbone path whose sparse structure is learned using the method we propose. To complement the the model's power of extracting features, narrow Skip-paths ($x - h_3$, $h_1 - h_3$, $h_2 - h_3$) are added to the model. The Skip-paths take all the

Table 1: Statistics of all the datasets.

| Dataset | Task | Classes | Training Samples | Validation Samples | Test Samples |
|---|---|---|---|---|---|
| Tox21 | Toxicity prediction | 2 | $6,901 \sim 9,154$ | 500 | $516 \sim 622$ |
| Yelp Review Full | Sentiment prediction | 5 | 640,000 | 10,000 | 50,000 |
| DBPedia | Topic classification | 14 | 549,990 | 10,010 | 70,000 |
| Sogou News | Topic classification | 5 | 440,000 | 10,000 | 60,000 |
| Yahoo!Answer | Topic classification | 10 | 1,390,000 | 10,000 | 60,000 |
| AG's News | Topic classification | 4 | 110,000 | 10,000 | 7,600 |

feature layers in the Backbone as input and compress them to layers with a small number of units through fully-connected projections.

## 3.3 BUILDING BSNN

After the structure for the Backbone path and the Skip-paths are determined, a classification layer or regression layer can then be added to the top of all the paths, utilizing all the features extracted. The network can then be trained using back-propagation algorithms as in normal neural networks.

## 4 EXPERIMENTS

In experiment, we evaluate our method in 17 classification tasks. We consider applications where the data is neither spacial nor sequential. Unlike CNNs or RNNs where the structure is designed to exploit spatial or sequential correlation, few effort has been put to learn the structure of feedfoward neural networks, which have highly redudant parameters and is prone to overfit. Our proposed method learns the structure of feedforward neural network from data. It significantly reduces the model complexity and parameters while achieving better or comparable classification performance, and leads to models which are more interpretable.

### 4.1 DATASETS

Table 1 gives a summary of all the datasets used in the experiment. We choose 12 tasks for chemical compounds classification and 5 tasks for text classification. All the datasets are published by previous researchers and are available to the public.

**Tox21 challenge dataset.** [1] There are about 12,000 environmental chemical compounds in the dataset, each represented as its chemical structure. The tasks are to predict 12 different toxic effects for the chemical compounds. We treat them as 12 binary classification tasks. We filter out sparse features which are present in fewer than 5% compounds, and rescale the remaining 1,644 features to zero mean and unit variance. The dataset contains a training set and a test set, and we randomly sample 500 compounds from training data to build the validation set. All the experiments are run for three times and we report the average AUC together with the standard deviations.

**Text classification datasets.** [2] We use 5 text classification datasets from Zhang et al. (2015). After removing stop words, the top 10,000 frequent words in each dataset are selected as the vocabulary respectively and each document is represented as bag-of-words over the vocabulary. The validation set is randomly sampled from the original training samples. We run all the experiments for three times and report the average classification accuracies with the standard deviations.

---

[1]https://github.com/bioinf-jku/SNNs
[2]https://github.com/zhangxiangxiao/Crepe

Table 2: Hyper-parameters for the structure of FNNs.

| Hyper-parameter | Values considered |
|---|---|
| Number of hidden units | {512, 1024, 2048} |
| Number of hidden layers | {1,2,3,4} |
| Network shape | {Rectangle, Conic} |

## 4.2 EXPERIMENT SETUP

We compare our model with standard feed-forward neural networks (FNNs) and sparse neural networks whose weak connections are pruned (Pruned FNNs) in the 17 classification tasks. The models involved in the experiment are as follows:

- **BSNN:** Backbone-Skippath Neural Network is the resulting model of our method PSA. For all the tasks, we keep only 5% of the connections in the Backbone path and limit the number of units in the narrow Skip-paths to 100.

- **FNN:** Feed-forward Neural Network is a standard fully-connected neural network. It is mainly composed of linear layers and activation functions. Each hidden unit is connected to all neurons in the previous layer. Information flows from low layers to high layers in a feed-forward manner.

- **Pruned FNN:** Pruned Feed-forward Neural Network is trained by using the method proposed in Han et al. (2015). We Firstly train a fully-connected FNN from scratch, and then prune out the weak connections with small absolute weight values. The pruned network is then retrained from the initial training phase by keeping the surviving weight parameters.

We learn the structure of BSNNs using PSA. The number of layers, the number of hidden units in each layer and the sparse connections between adjacent layers are automatically determined. After structure learning, we train the sparse model from scratch by random initialization of weights. As for FNNs, we treat the number of hidden units and number of layers as hyper-parameters of network and determine the best structure by grid-search over all the combinations using validation data. Table 2 shows the space of network structures considered. Following the method in Klambauer et al. (2017) , both "rectangle " and "conic" network shapes are tested. In FNNs with rectangle shape, all the hidden layers have constant number of units. FNNs with conic shape start with the given number of hidden units and decrease it layer by layer in a geometric progression manner towards the output layer. For Pruned FNNs, we take the best FNNs as the initial model and perform pruning as in Han et al. (2015). The pruned model is then retrained for final model.

We implement all the experiments using PyTroch[3] which is a flexible deep learning framework. We use ReLUs (Nair & Hinton, 2010; Glorot et al., 2011) as the non-linear activation functions in all the networks. Dropout (Hinton et al., 2012b; Srivastava et al., 2014) with rate 0.5 is applied after each non-linear projection. We use Adam (Kingma & Ba, 2014) as the optimizer to optimize the training objective function. During training, we select models by monitoring validation loss. Codes will be released after the paper is accepted to the conference.

## 4.3 RESULTS

### 4.3.1 BSNNS VS FNNS

Table 3 shows the classification results of BSNNs and FNNs on Tox21 dataset. The structures of FNNs are tuned individually for each task. It is clear that BSNNs achieve better AUC scores on 10 out of the 12 classification tasks. Even when it is not better, the average AUC value of BSNNs, e.g. on task *SR.MMP*, is also very close to that of FNNs. More importantly, BSNNs always contain much fewer parameters than FNNs, with the ratios of parameter number ranging from $7\%$ to $40.11\%$.

Table 4 shows the results of BSNNs and FNNs over the 5 text classification tasks. Although BSNNs contain much fewer parameters than FNNs, BSNNs still achieve higher classification accuracy in

---

[3]https://github.com/pytorch/pytorch

Table 3: Comparison between BSNNs and FNNs on Tox21 challenge dataset. The structures of FNNs are chosen by using validation data. Each experiment is run for three times.

| Task | BSNNs | | | FNNs | |
|------|-------|--|--|------|--|
| | AUC | Parameter # / Ratio w.r.t FNNs | | AUC | Parameter # |
| NR.AhR | $0.8930 \pm 0.0014$ | 338K / | 37.08% | $0.8843 \pm 0.0030$ | 912K |
| NR.AR | $0.7316 \pm 0.0245$ | 338K / | 20.05% | $0.6629 \pm 0.0155$ | 1.69M |
| NR.AR.LBD | $0.7827 \pm 0.0200$ | 338K / | 20.05% | $0.7216 \pm 0.0245$ | 1.69M |
| NR.Aromatase | $0.7854 \pm 0.0098$ | 338K / | 40.11% | $0.7834 \pm 0.0046$ | 843K |
| NR.ER | $0.7804 \pm 0.0042$ | 338K / | 12.36% | $0.7671 \pm 0.0090$ | 2.73M |
| NR.ER.LBD | $0.7772 \pm 0.0088$ | 338K / | 20.75% | $0.8145 \pm 0.0035$ | 1.63M |
| NR.PPAR.gamma | $0.8232 \pm 0.0019$ | 338K / | 39.38% | $0.8024 \pm 0.0098$ | 858K |
| SR.ARE | $0.7877 \pm 0.0036$ | 338K / | 7.00% | $0.7809 \pm 0.0092$ | 4.83M |
| SR.ATAD5 | $0.8188 \pm 0.0085$ | 338K / | 40.11% | $0.7980 \pm 0.0014$ | 843K |
| SR.HSE | $0.8330 \pm 0.0053$ | 338K / | 30.59% | $0.8318 \pm 0.0047$ | 1.10M |
| SR.MMP | $0.9249 \pm 0.0014$ | 338K / | 20.05% | $0.9253 \pm 0.0038$ | 1.69M |
| SR.p53 | $0.8425 \pm 0.0023$ | 338K / | 40.11% | $0.8401 \pm 0.0049$ | 843K |
| Average | $0.8150 \pm 0.0038$ | | 27.30% | $0.8010 \pm 0.0017$ | |

Table 4: Comparison between BSNNs and FNNs on 5 text classification datasets. The structures of FNNs are chosen by using validation data. Each experiment is run for three times.

| Task | BSNNs | | | FNNs | |
|------|-------|--|--|------|--|
| | Accuracy | Parameter # / Ratio w.r.t FNNs | | Accuracy | Parameter # |
| Yelp Review Full | $59.14\% \pm 0.06\%$ | 1.73M / | 32.07% | $59.13\% \pm 0.14\%$ | 5.38M |
| DBPedia | $98.11\% \pm 0.03\%$ | 1.78M / | 17.13% | $97.99\% \pm 0.04\%$ | 10.36M |
| Sogou News | $96.09\% \pm 0.06\%$ | 1.84M / | 13.77% | $96.12\% \pm 0.06\%$ | 13.39M |
| Yahoo!Answer | $71.42\% \pm 0.06\%$ | 1.69M / | 31.42% | $71.84\% \pm 0.07\%$ | 5.39M |
| AG's News | $91.39\% \pm 0.03\%$ | 1.81M / | 6.25% | $91.61\% \pm 0.01\%$ | 28.88M |

the first two tasks, and comparable accuracy in the remaining tasks. Note that the ratios of parameter number ranges from 6.25% to 32.07%. This again confirms that our method learns good parsimonious deep models which can achieve high classification performance with much fewer parameters than standard FNNs.

### 4.3.2 CONTRIBUTION OF THE BACKBONE

Table 5: Comparison between BSNNs and BSNNs with only the backbone path. Tox21 Average corresponds to the result averaged over the 12 tasks in Tox21 dataset. Each experiment is run for three times.

| Task | BSNNs | Backbone Path in BSNNs | | |
|------|-------|------------------------|--|--|
| | AUC/Accuracy | AUC/Accuracy | Parameter ratio w.r.t BSNNs | Parameter ratio w.r.t FNNs |
| Tox21 Average | $0.8150 \pm 0.0038$ | $0.7839 \pm 0.0076$ | 30.47% | 8.32% |
| Yelp Review Full | $59.14\% \pm 0.06\%$ | $58.63\% \pm 0.13\%$ | 35.47% | 11.38% |
| DBPedia | $98.11\% \pm 0.03\%$ | $97.91\% \pm 0.04\%$ | 36.67% | 6.28% |
| Sogou News | $96.09\% \pm 0.06\%$ | $95.67\% \pm 0.04\%$ | 38.63% | 5.32% |
| Yahoo!Answer | $71.42\% \pm 0.06\%$ | $69.95\% \pm 0.08\%$ | 34.39% | 10.80% |
| AG's News | $91.39\% \pm 0.03\%$ | $91.33\% \pm 0.03\%$ | 37.55% | 2.35% |

To validate our assumption that the backbone path in BSNNs captures most of the information in data and acts as a main part of the model, we remove the narrow skip-paths in BSNNs and train

the model to test its performance in classification tasks. Table 5 shows the results. As we can see from the results, the backbone path alone already achieves AUC scores or accuracies which are only slightly worse than BSNNs. Note that the number of parameters in the sparse path is even much smaller than BSNNs. Compared with FNNs, the number of parameters is only 2% 11%, significantly smaller than that of FNNs. However, without the backbone, the performance of the model will be significantly worse due to the insufficient capability of the other narrow path. The results not only show the importance of the backbone path in BSNNs, but also shows that our structure learning method in the backbone path is effective enough.

Table 6: AUC scores of BSNNs, BSNN-FCs and Pruned FNNs on Tox21 dataset. For each task, better result between BSNNs and BSNN-FCs is underlined, while better result between BSNNs and Pruned FNNs is **bold**.

| Task | BSNNs | BSNN-FCs | Pruned FNNs |
|---|---|---|---|
| NR.AhR | **0.8930 ± 0.0014** | 0.8910 ± 0.0014 | 0.8845 ± 0.0047 |
| NR.AR | **0.7316 ± 0.0245** | 0.6780 ± 0.0252 | 0.6660 ± 0.0206 |
| NR.AR.LBD | **0.7827 ± 0.0200** | 0.7796 ± 0.0136 | 0.7475 ± 0.0356 |
| NR.Aromatase | **0.7854 ± 0.0098** | 0.7757 ± 0.0124 | 0.7782 ± 0.0069 |
| NR.ER | **0.7804 ± 0.0042** | 0.7693 ± 0.0049 | 0.7767 ± 0.0059 |
| NR.ER.LBD | 0.7772 ± 0.0088 | 0.7970 ± 0.0057 | **0.8054 ± 0.0071** |
| NR.PPAR.gamma | **0.8232 ± 0.0019** | 0.8136 ± 0.0032 | 0.7803 ± 0.0045 |
| SR.ARE | **0.7877 ± 0.0036** | 0.7771 ± 0.0058 | 0.7812 ± 0.0024 |
| SR.ATAD5 | **0.8188 ± 0.0085** | 0.8162 ± 0.0062 | 0.7924 ± 0.0051 |
| SR.HSE | **0.8330 ± 0.0053** | 0.8453 ± 0.0072 | 0.8308 ± 0.0103 |
| SR.MMP | 0.9249 ± 0.0014 | 0.9219 ± 0.0004 | **0.9262 ± 0.0036** |
| SR.p53 | **0.8425 ± 0.0023** | 0.8194 ± 0.0010 | 0.8278 ± 0.0090 |
| | | | |
| Average | **0.8150 ± 0.0038** | 0.8070 ± 0.0002 | 0.7998 ± 0.0034 |

### 4.3.3 EFFECTIVENESS OF OUR STRUCTURE LEARNING

To further show the effectiveness of our structure learning method, we introduce a new model called BSNN-FC. For each specific task, the structure of BSNN-FC is completely the same as that of BSNN, except that the layers in the sparse Backbone path are changed to fully-connected layers. We train BSNN-FC for all the tasks in Tox21 dataset and the results are shown in Table 6. From the table we can see that, although BSNN keeps only 5% of the connections in the sparse path, it gives classification results which are very similar to that of BSNN-FC. It shows that our structure learning method successfully removes the useless connections in BSNN-FC.

We also compare BSNNs with Pruned FNNs whose weak connections are pruned using the method in Han et al. (2015). We start from the fully pretrained FNNs reported in Table 3, and prune the connections with the smallest absolute weight values. After pruning, the number of remaining parameters in each FNN is the same as that in the corresponding BSNN for the same task. The comparison between BSNNs and pruned FNNs is shown in Table 6. Again BSNNs give higher AUC scores than pruned FNNs in 10 of the 12 classification tasks.

### 4.3.4 INTERPRETABILITY

Next we compare the interpretability of BSNNs with FNNs and Pruned FNNs on the text datasets. Here is how we interpret hidden units. We feed the data to the networks and do forward propagation to get the values of the hidden units corresponding to each data sample. Then for each hidden unit, we sort the words in descending order of the correlations between the words and the hidden unit. The top 10 words with the highest correlations are chosen to characterize the hidden unit. Following Chen et al. (2017b), we measure the interpretability of a hidden unit by considering how similar pairs of words in the top-10 list are. The similarity between two words is determined using a word2vec model (Mikolov et al., 2013a;b) trained on part of the Google News datasets, where each word is mapped to a high dimensional vector. The similarity between two words is defined as the cosine similarity of the two corresponding vectors. High similarity suggests that the two words

Table 7: Interpretability scores of BSNNS, FNNs and Pruned FNNs on different datasets

| Task | BSNNs | FNNs | Pruned FNNs |
|------|-------|------|-------------|
| Yelp Review Full | **0.1632** | 0.1117 | 0.1 |
| DBPedia | **0.0609** | 0.0497 | 0.0553 |
| Yahoo!Answer | **0.1729** | 0.1632 | 0.1553 |
| AG's News | 0.0531 | **0.0595** | 0.0561 |

Table 8: Qualitative interpretability results of hidden units in BSNNs. Each line corresponds to one hidden unit.

| Task | BSNNs |
|------|-------|
| Yelp Review Full | tasteless unseasoned flavorless bland lacked
paprika panko crusts unagi crumb
vindaloo tortas spicey wink drapes |
| DBPedia | album songwriting chet saxophone thrash
hurling backstroke badminton skier outfielder
journalists hardcover editors reprinted republished |
| Yahoo!Answer | harddrive antispyware wifi mcafee routers
javascript linux tcp linksys laptops
romantic dating foreplay flirt boyfriend |
| AG's News | mozilla mainframe designs collaborate microprocessors
republicans prosecutor argument jfk protesters
noted furious harsh concessions apologizes |

appear in similar contexts. The interpretability score of a hidden unit is defined as the compactness of its characterizing words and is computed as the average similarity of all pairs of words. The interpretability score of a model is defined as the average of interpretability scores of all hidden units.

Table 7 reports the interpretability scores of BSNNs, FNNs and Pruned FNNs for different datasets. *Sogounews* dataset is not included in the experiment since its vocabulary are Chinese pingyin characters and most of them do not appear in the Google News word2vec model. We measure the interpretability scores by considering the top-layer hidden units. For the fair of comparison, all models have approximately the same number of top-layer hidden units. As it can be seen that BSNNs significantly outperform the FNNs and Pruned FNNs in most cases and is comparable if not better, showing superior coherency and compactness in the characterizations of the hidden units and thus better model interpretability. Pruned FNNs, on the other hand, reduce the interpretability of FNNs with the pruning strategy. Table 8 shows the qualitative interpretability results by presenting the characterization words of hidden units with high interpretability scores in BSNNs. The hidden units are very meaningful for different datasets. For example, in *Yelp Review* dataset, the first hidden unit represents negative opinions on food with words "tasteless" and"flavorless"; the second hidden unit is more related to food like "paprika", "crust" and "unagi". In *DBPedia*, the first hidden unit is found out to have closer relationship with music, while the second one is more closely related to sport. Similar phenomena can be found in the rest of the table. This shows that the proposed BSNNs, with the statistical property, have better model interpretability and make a step further towards understandable deep learning models.

## 5 CONCLUSIONS

Structure learning for deep neural network is a challenging and interesting research problem. We have proposed an unsupervised structure learning method which utilizes the correlation information in data for learning parsimonious deep feed-forward networks. In comparison with standard FNN, although the resulting model of our method contains much fewer parameters, it achieves better or comparable classification performance in all kinds of tasks. Our method is also shown to learn

models with better interpretability, which is also an important problem in deep learning. In the future, we will generalize our method to other networks like RNNs and CNNs.

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
