# OpenReview forum: "Learning Parsimonious Deep Feed-forward Networks"
_ICLR.cc/2018/Conference — Reject_

### Official Review · AnonReviewer1 · 2017-11-26
**Learning Parsimonious Deep Feed-forward Networks**

**Rating:** 5
**Confidence:** 5

**Review:**

There is a vast literature on structure learning for constructing neural networks (topologies, layers, learning rates, etc.) in an automatic fashion. Your work falls under a similar category. I am a bit surprised that you have not discussed it in the paper not to mention provided a baseline to compare your method to. Also, without knowing intricate details about each of 17 tasks you mentioned it is really hard to make any judgement as to how significant is improvement coming from your approach. There has been some work done on constructing interpretable neural networks, such as stimulated training in speech recognition, unfortunately these are not discussed in the paper despite interpretability being considered important in this paper.

---

> ### Author Response · Authors · 2017-12-05
> **RE: AnonReviewer1's review**
>
> Thank you for your reviews.
>
> #Discussion of literature on structure learning for neural networks is missing#
> No. We Do have the discussion covering most of the important methods, e.g. constructive algorithm (Ash, 1989; Bello, 1992; Kwok & Yeung, 1997), RL algorithm, Genetic algorithm, pruning and so on in our Related Works section. Please take a look at it. And we also compare with a baseline method (pruning) in our experiments.
>
> #Unclear tasks and unclear improvement#
> No. Firstly, text classification is well studied in the literature and is not at all a mysterious task. In addition, the five large-scale text datasets we included are among the most important text classification datasets nowadays. The Tox21 dataset is also studied in a famous NIPS2017 paper, Self-Normalizing Neural Networks (SELU), in a similar setting. Secondly, we want to emphasize that the goal of this paper is NOT to propose state-of-the-art solutions to the 17 classification tasks, but to propose a structure learning method and compare it with baselines on the 17 tasks. Last but not the least, even when all the baseline FNN structures are fully tuned over the validation data, our method still achieves better/comparable classification performances in all the 17 tasks. This is a clear validation of the effectiveness of our structure learning method, considering the Backbone path in our model contains only 5% of the connections.
>
> #Paper on interpretable neural networks are not discussed#
> The goal of this paper is to propose a structure learning method for *Parsimonious* neural networks such that the models contain fewer parameters than standard FNNs but still achieve better performance in different tasks. The method is not directly optimizing the structures for interpretability. Better interpretability (than baselines) is just one resulting advantage of our method and hence we think it is not necessary to include a heavy discussion on papers about interpretable neural networks. If the reviewer think that it is necessary, we will add it in our revision.

---

> > ### Comment · AnonReviewer1 · 2017-12-23
> > **Learning Parsimonious Deep Feed-forward Networks**
> >
> > My confusion with your point #1 is a simple fact that you are proposing a method of constructing a NN using some form of a cost function. There is a lot of literature where people are trying to build NN using target evaluation metric as the cost function. For classification tasks this would be classification accuracy. Building NN here consist of adding/removing layers, changing learning rates, etc. These are so called architecture search methods. I am aware that these methods are more expensive yet they attempt to come up with a custom architecture for given problem like your method does. As such I expected to see more discussion in this directions.
> >
> > My confusion with your point#2 stems from you claiming to provide validation on 17 different tasks,
> > 12 out of these 17 tasks come from Tox21 data set. Let us look at table 3, what are NR.AhR, NR.AR, ..., SR.p53 tasks, how important is improvement on NR.AhR, is improvement on NR.AR more important than it is on  NR.AhR, how significant is the difference between 0.8930 and 0.8843, what is state-of-the art on each of these sets (for feedforward, also other models), is this really correct that BSNN has 338K on all these tasks. For Table 4 similarly, what is state-of-the-art here?
> >
> > You treat interpretability rather seriously in this paper so I do think you need to refer to other work done in that area. Second of all, given the way you treat interpretability I would expect you conducting some subjective evaluations by asking human subject to rank models based on the way they group words. I find it hard to be convinced given similarity values such as 0.1729, 0.1632, 0.1553 you compute by means of embeddings derived from word2vec model.

---

> > > ### Author Response · Authors · 2018-01-04
> > > **RE: AnonReviewer1's review**
> > >
> > > Thank you for your explanations.
> > >
> > > #1
> > > No. We are NOT “using some form of a cost function”, but proposing an unsupervised learning method. If our understanding is correct, the reviewer is talking about methods of manually validating network structure over validation data. Note that all the baseline FNNs in our experiments are validated over validation data. If the reviewer thinks it necessary to have more discussion in that directions, we will include it.
> > >
> > > #2
> > > We agree that more introduction and references for Tox21 dataset would help reader better understand the experiment results. Thank you for your suggestions.
> > >
> > > #3
> > > Thank you for your suggestions.

---

### Official Review · AnonReviewer3 · 2017-11-28
**Learning Parsimonious Deep Feed-forward Networks**

**Rating:** 4
**Confidence:** 2

**Review:**

This paper introduces a skip-connection based design of fully connected networks, which is loosely based on learning latent variable tree structure learning via mutual information criteria. The goal is to learn sparse structures across layers of fully connected networks.  Compared to prior work (hierarchical latent tree model), this work introduces skip-paths.
Authors refer to prior work for methods to learn this backbone model. Liu et.al (http://www.cse.ust.hk/~lzhang/ltm/index.htm) and Chen et.al. (https://arxiv.org/abs/1508.00973) and (https://arxiv.org/pdf/1605.06650.pdf).

As far as I understand, the methods for learning backbone structure and the skip-path are performed independently, i.e. there is no end-to-end training of the structure and parameters of the layers. This will limit the applicability of the approach in most applications where fully connected networks are currently used.

Originality - The paper heavily builds upon prior work on hierarchical latent tree analysis and adds 'skip path' formulation to the architecture, however the structure learning is not performed end-to-end and in conjunction with the parameters.

Clarity - The paper is not self-contained in terms of methodology.

Quality and Significance - There is a disconnect between premise of the paper (improving efficiency of fully connected layers by learning sparser structures) and applicability of the approach (slow EM based method to learn structure first, then learn the parameters).  As is, the applicability of the method is limited.
Also in terms of experiments, there is not enough exploration of simpler sparse learning methods such as heavy regularization of the weights.

---

> ### Author Response · Authors · 2017-12-05
> **RE: AnonReviewer3's review**
>
> Thank you for your reviews.
>
> #End-to-end training#
> We wish to remind the reviewer that we are proposing an *Unsupervised* structure learning method. One key advantage of unsupervised structure learning is that it can make use of both unlabelled and labelled data, and the learned structure can be transferred to any tasks on the same type of data. Think about the structure of convolutional layer which is used across all kinds of CV tasks. Why don't we train the connectivities of convolutional layer with the parameters in an end-to-end fashion? The reason is that we humans have seen many unlabelled scenes, we know a strong pattern in vision data and hence we design a specific structure suited to that pattern without further learning. Similarly, our method is trying to find such strong patterns in general data other than images and build structures correspondingly, followed by parameter learning in specific tasks. If you train the structure and parameters in an end-to-end manner, then it is supervised learning and task-specific, which is not what we want.
>
> In addition, compared with an end-to-end method (pruning), our method has achieved higher classification AUC scores in 10 out of 12 tasks and significantly higher interpretability scores in 3 out of 4 tasks. It is clear that the end-to-end method shows no superiority to our method.
>
> #Originality#
> We want to emphasize the contributions of our paper. Note that prior works on hierarchical latent tree analysis are proposing structure learning methods for Bayesian network, while in this paper we aim at structure learning of deep feed-forward neural networks.
> 1. It is the first time that the latent tree-based structure learning method is applied to multi-layer neural network and supervised learning task (classification). Previous works on such topic are for unsupervised tasks only.
> 2. This paper proposes a method for learning multi-layer deep sparse feed-forward neural network. This is different from previous works in that previous works on latent tree model learn either multi-layer tree model (Chen et al. 2017a) or two-layer sparse model Chen et al. (2017b).
>
> #Inefficient due to slow EM algorithm.#
> No. Firstly, we use *Progressive EM* (Chen et al., 2016) and *Stepwise EM* (similar to SGD) (Sato and Ishii 2000; Cappe and Moulines 2009) in our method. They have been shown to be efficient and can easily scale up for hundreds of thousands of training samples in previous works. Secondly, structure learning is only needed during offline training, and the learned sparse connections can speed up online testing. Besides, our method is proposed not only for efficiency, but also for model fit and model storage.
>
> #Regularization of the weights as baseline are missing#
> No. The pruning method we compare with is usually regarded as a strong regularization over weights in the literature. The regularization is even stronger than l1 norm as it is producing many weights being exactly 0.

---

### Official Review · AnonReviewer4 · 2017-12-06
**Needs improvement**

**Rating:** 5
**Confidence:** 2

**Review:**

The main strengths of the paper are the supporting experimental results in comparison to plain feed-forward networks (FNNs).  The proposed method is focused on discovering sparse neural networks.  The experiments show that sparsity is achieved and still the discovered sparse networks have comparable or better performance compared to dense networks.

The main weakness of the paper is lack of cohesion in contributions and difficulty in delineating the scope of their proposed approach.

Below are some suggestions for improving the paper:

Can you enumerate the paper’s contributions and specify the scope of this work?  Where is this method most applicable and where is it not applicable?

Why is the paper focused on these specific contributions?  What problem does this particular set of contributions solve that is not solvable by the baselines?  There needs to be a cohesive story that puts the elements together.  For example, you explain how the algorithm for creating the backbone can use unsupervised data.  On the other hand, to distinguish this work from the baselines you mention that this work is the first to apply the method to supervised learning problems.

The motivation section in the beginning of the paper motivates using the backbone structure to get a sparse network.  However, it does not adequately motivate the skip-path connections or applications of the method to supervised tasks.

Is this work extending the applicability of baselines to new types of problems?  Or is this work focused on improving the performance of existing methods?  Answers to these questions can automatically determine suitable experiments to run as well.  It's not clear if Pruned FNNs are the most suitable baseline for evaluating the results.  Can your work be compared experimentally with any of the constructive methods from the related work section?  If not, why?

When contrasting this work with existing approaches, can you explain how existing work builds toward the same solution that you are focusing on?  It would be more informative to explain how the baselines contribute to the solution instead of just citing them and highlighting their differences.

Regarding the experimental results, is there any insight on why the dense networks are falling short?  For example, if it is due to overfitting, is there a correlation between performance and size of FNNs?  Do you observe a similar performance vs FNNs in existing methods?  Whether this good performance is due to your contributions or due to effectiveness of the baseline algorithm, proper analysis and discussion is required and counts as useful research contribution.

---

> ### Author Response · Authors · 2018-01-04
> **RE: AnonReviewer4's review**
>
> Thank you for your suggestions.

---

### Decision · Program_Chairs · 2018-01-29
**ICLR 2018 Conference Acceptance Decision**

**Decision:**

Reject

**Comment:**

I am inclined to agree with R1 that there is an extensive literature on learning architectures now, and I have seen two others as part of my area chairing. This paper does not offer comparisons to existing methods for architecture learning other than very basic ones and that reduces the strength of the paper significantly. Further the broad exploration over 17 tasks is more overwhelming, than adding to an insight into the methods.